# Genome-Wide Analysis of the MADS-Box Gene Family in *Hibiscus syriacus* and Their Role in Floral Organ Development

**DOI:** 10.3390/ijms25010406

**Published:** 2023-12-28

**Authors:** Jie Wang, Heng Ye, Xiaolong Li, Xue Lv, Jiaqi Lou, Yulu Chen, Shuhan Yu, Lu Zhang

**Affiliations:** 1College of Landscape Architecture, Zhejiang A&F University, Hangzhou 311300, China; jiewang2142022@163.com (J.W.); 2021105051004@stu.zafu.edu.cn (H.Y.); mizaiaimimi@163.com (X.L.); belieflou@163.com (J.L.); yuluchen-1@outlook.com (Y.C.); 2College of Horticulture Science, Zhejiang A&F University, Hangzhou 311300, China; lixiaolong@zafu.edu.cn

**Keywords:** *H. syriacus*, MADS-box, genome-wide characterization, flower organ development, expression profiling

## Abstract

*Hibiscus syriacus* belongs to the Malvaceae family, and is a plant with medicinal, edible, and greening values. MADS-box transcription factor is a large family of regulatory factors involved in a variety of biological processes in plants. Here, we performed a genome-wide characterization of MADS-box proteins in *H. syriacus* and investigated gene structure, phylogenetics, *cis*-acting elements, three-dimensional structure, gene expression, and protein interaction to identify candidate MADS-box genes that mediate petal developmental regulation in *H. syriacus*. A total of 163 candidate MADS-box genes were found and classified into type I (Mα, Mβ, and Mγ) and type II (MIKC and M*δ*). Analysis of *cis*-acting elements in the promoter region showed that most elements were correlated to plant hormones. The analysis of nine *HsMADS* expressions of two different *H. syriacus* cultivars showed that they were differentially expressed between two type flowers. The analysis of protein interaction networks also indicated that MADS proteins played a crucial role in floral organ identification, inflorescence and fruit development, and flowering time. This research is the first to analyze the MADS-box family of *H. syriacus* and provides an important reference for further study of the biological functions of the MADS-box, especially in flower organ development.

## 1. Introduction

The MADS-box transcription factor family contains a conserved motif with 60 amino acids [1,2]. MADS-box genes are classified into two groups, type I and type II, which originate from the replication of ancestral genes. Type II genes seem to evolve slower than type I [3], which belong to a heterogeneous group with only about 180 bp nucleotide sequences that together encode the MADS domain [4,5,6]. Type I MADS-box genes can be further classified into Mα, Mβ, and Mγ subfamilies [7]. The type II MADS-box proteins include a modular domain structure, which is named the MIKC structure. Type II MADS-box genes include the MEF2-type, which is found in animals and fungi, and the MIKC-type, which is present in plants [8]. Four characteristic domains exist in MIKC-type proteins, which are named MADS-box (M), intermediate (I), keratin-like (K), and C-terminal (C) domains. It is important to note that the K domain is missing in the type I MADS domain protein [9]. The most conservative of the four domains is the M domain, followed by I and K domains, while the C domain is the least conserved and contains a transactivation structure. The I domain facilitates dimerization specificity, and the K domain typically features a coil structure [5].

MADS-box genes play a critical role in various aspects of plant reproductive development, especially in seed development, inflorescence architecture, and floral organ identity [10]. Many transcription factors (TFs) controlling flower development in model plants have been identified and constitute an “ABC model” [11], which is then extended to the “ABCDE model” [12]. Most genes in the “ABCDE model” belong to MADS-box transcription factors, which are the basis for controlling floral development [3,7,13]. According to the “ABCDE” model, class A genes and class E genes control the growth of sepals in the first round of floral organs. Class A genes, class B genes, and class E genes are jointly involved in the formation of petals in the second round. In the third round, class B genes, class C genes, and class E genes jointly regulate the development of stamens. In the fourth round, the formation of the pistil is controlled by class C genes and class E genes, and class D genes act on ovule development [14,15]. To date, the MADS-box genes family in many plants have been analyzed [16]. For instance, the MADS-box genes of *Ananas comosus* are associated with flower and fruit development [17]. In *Zea mays*, the type I MADS-box genes have a faster birth and death rate compared to the type II MADS-box genes [18]. In *Glycine max*, the expression of type II MADS-box genes is much higher during the development of seeds and flower buds than that of type I, suggesting that type II MADS-box genes play more essential roles than type I genes in these processes [19]. The MADS-box genes of *Alcea rosea* are related to the stamen petaloid by regulating plant hormones [20]. Predicting the three-dimensional (3D) structure of proteins and protein interactions plays a crucial role in predicting the protein function of target proteins and analyzing signal transduction pathways [21]. However, the current analysis of the MADS-box gene family limitedly involves the examination of protein 3D structures and protein interactions.

*Hibiscus syriacus* (2*n* = 4*x* = 80) belongs to the Malvaceae family and is a polyploid species [22]. It has high ornamental, medicinal, and edible values [23]. The genome sequencing of *H. syriacus* have been accomplished, and the whole genome comprises a 1.75 Gb assembly that covers 92% of the genome with only 1.7% (33 Mb) gap sequences [22]. The flowers of *H. syriacus* have several types, including single, semi-double, and double petals [24]. Double-flower formation in *Kerria japonica* can be attributed to the MADS-box gene, which appears to be a mutant produced by the insertion of a transposon-like fragment in the normal *AGAMOUS* (*AG*) homolog, which controls the development of the stamen and pistil of the single flower. The loss of *KjAG* function not only affects its own expression, but also affects other putative floral organ identity genes [25]. The double flower of *Prunus lannesiana* is also caused by the inhibition of AG-like gene expression in whorl 3 [26]. However, the molecular and genetic mechanisms of the floral development of *H. syriacus* have not been deeply studied. With the accomplishment of the genome sequencing of *H. syriacus*, it is feasible and necessary to analyze the MADS-box gene family in the whole genome of *H. syriacus* [22].

In this study, we identified 163 MADS-box genes in total in *H. syriacus*. The gene structure, phylogenetic relationship, multiple sequences alignment, conservative motif, 3D structure, *cis*-acting elements, expression patterns, and protein interaction of MADS-box genes in *H. syriacus* were analyzed. The result will provide a theoretical basis for breeding *H. syriacus* with different floral types.

## 2. Result

### 2.1. Identification and Classification of the HsMADS-Box Genes

Using the *Arabidopsis* MADS-box genes as query sequences, *MADS-box* genes in the whole genome of *H. syriacus var.* Baekdansim were searched for and a total of 277 candidate genes were identified. Then these candidate genes were further verified and screened based on a structural domain using NCBI Batch CD-search. A total of 163 *HsMADS-box* genes were identified. According to the ID sequence of the MADS-box gene in the NCBI annotation information, *HsMADS-box* genes were named from *HsMADS1* to *HsMADS163*. The gene ID, molecular weight (MW), protein sequence length (aa), and isoelectric point (pI) are shown in Appendix A. The MW of the HsMADS-box proteins ranged from 11.44 kDa to 48.87 kDa, with an average of 26.15 kDa. The length of the HsMADS-box proteins ranged from 100 (*HsMADS134*) to 447 (*HsMADS131*), with an average of 229. The PI ranged from 4.88 to 10.18, with an average of 8.67.

### 2.2. Multiple Sequence Alignment and Phylogenetic Relationships of the HsMADS-Box Genes

The MADS domain of HsMADS-box proteins was aligned (Figure 1), and contains approximately 60 amino acids and is present in all MADS-box genes [27]. In the MIKC, M*δ*, Mβ, and Mγ subfamilies, the MADS domain of most genes was highly conservative. However, in the Mα subfamily, only five genes out of seventeen genes were highly conserved (*HsMADS85*, *HsMADS90*, *HsMADS98*, *HsMADS139*, and *HsMADS147*), indicating that this subfamily had evolved faster and was prone to mutate. All sequences were shown in Appendix A.

To gain a clearer understanding of the evolutionary relationships of MADS-box family members in *H. syriacus*, the maximum likelihood (ML) method was used to construct a phylogenetic tree of MADS-box members from *H. syriacus* (163), *A. thaliana* (102), and *G. raimondii* (199) (Figure 2a). *G. raimondii* was selected because it has a high-quality reference genome and a close relationship with *H. syriacus*. MADS-box genes were classified into type I and type II. The number of MADS-box genes in type II (134) was significantly more than that in type I (29). Type I MADS-box genes were classified into Mα, Mβ, and Mγ subfamilies. Both Mβ and Mγ subfamilies included six genes, and the Mα subfamily had the most members (seventeen genes). Type II MADS-box genes were classified into M*δ* and MIKC subfamilies, which had 4 and 130 genes, respectively. Type II genes were more closely related to *A. thaliana* than type I, indicating that type I evolved faster than type II.

### 2.3. Conservative Motif Distribution and Gene Structure Analysis of HsMADS-Box Genes

The intron/exon arrangement and conserved motifs based on phylogenetic relationships were analyzed. The results showed that HsMADSs contains 0–11 exons. Most type II genes included multiple introns, while type I genes had no or only one intron (Figure 3). Among the eight MADS-box genes with one exon, only HsMADS134 belonged to the type II gene category. There were less than eight introns in most *H. syriacus* MADS-box genes. Only *HsMADS24*, *HsMADS136*, and *HsMADS162* belonging to the M*δ* subfamily contained more introns, i.e., 9, 10, and 11, respectively. Genes that had close relationships such as *HsMADS6*, *HsMADS7*, *HsMADS8*, and *HsMADS9* generally had similar gene structures. However, the number of introns might be significant difference between the closely related genes; for example, *HsMADS134* had only one intron, but *HsMADS143* had seven introns (Figure 3).

A total of ten motifs were identified, which were named motifs 1–10 (Figure 4). Motif 1, motif 4, and motif 6 were considered to be the MADS structural domain. Almost all of the MADS-box genes (except for *HsMADS47*) contained motif 1, which existed in the N-terminus. Motif 2 represented the K domain and existed in all MIKC proteins except *HsMADS16*, *HsMADS44*, *HsMADS101*, *HsMADS134*, and *HsMADS146*, while it only existed in two type I genes (*HsMADS47* and *HsMADS103*). Motif 3 was considered to be the I domain, which mainly existed in type II genes, except *HsMADS11*. Compared with other motifs, motif 1 and motif 4 were relatively conservative.

The evolutionary trajectory of a protein through sequence space is constrained by its function. The 3D protein fold with remarkable accuracy can infer evolutionary from a set of sequence homologs of a protein [28]. We selected two MADS-box genes from each subfamily and used the Swiss-Model to predict the 3D structure of homologous model of HsMADS proteins (Figure 5). The results showed that the Root Mean Square Displacement (RMSD) values of HsMADS proteins were all less than 1, indicating that the results were reliable and HsMADS proteins have similar structures. HsMADS12 and HsMADS161 had the most similar 3D structure (RMSD = 0.035). Most MADS-box proteins had four α-helix and four β-pleated sheets; only HsMADS58, HsMADS11, HsMADS87, and HsMADS24 had six α-heli. The results highlighted that the HsMADS proteins were highly conserved in structure.

### 2.4. Analysis of cis-Acting Elements in the Promoter Region of HsMADS-Box Genes

The *cis*-acting elements at 2000 bp upstream from the transcription initiation site was analyzed in MADS-box genes. A total of 95 *cis*-acting elements were found, and 10 important *cis*-acting elements were selected, and were involved in the abscisic acid response (ABRE), MeJA-responsive (CGTCA-motif and TGACG-motif), gibberellin-responsiveness (TATC-box), meristem-specific activation (NON-box), auxin-responsive element (TGA-element), zein metabolism regulation (O_2_-site), part of an auxin-responsive element (TGA-box), gibberellin-responsive element (P-box and GARE-motif), meristem expression (CAT-box), and auxin responsiveness (AuxRR-core) (Figure 6a). A total of 352 abscisic acid response elements, 71 zeatin response elements, 21 meristem response elements, 71 gibberellin response elements, 424 jasmonate response elements, and 121 auxin response elements were identified in all *HsMADS-box* genes (Figure 6b). In addition, abscisic acid-responsive element, MeJA-responsive element, gibberellin-responsive element, and auxin-responsive element generally existed in MADS-box genes, while meristem response elements mainly existed in type II genes.

### 2.5. Expression Analysis of MADS-Box Genes in Six H. syriacus Cultivars

The expression of nine MADS-box genes, whose Log2 FC > 1.5 and FPKM values were higher than 0.5, were analyzed based on transcriptome data from flower buds of two *H. syriacus* cultivars in two different developmental stages. The results suggested that the expression of *HsMADS48*, *HsMADS69*, *HsMADS111*, and *HsMADS161* in double flowers was significantly lower, while the expression of *HsMADS8*, *HsMADS33*, and *HsMADS145* was significantly higher compared with that of single flowers (Figure 7).

In order to verify the role of MADS genes in floral organ development, nine significantly differentially expressed MADS-box genes were selected to conduct qRT-PCR analysis. The results were generally consistent with transcriptome results. In the same period, the expression of *HsMADS48*, *HsMADS69*, *HsMADS111*, *HsMADS138*, *HsMADS143* and *HsMADS161* was significantly lower in double flowers, while the expression of *HsMADS8*, *HsMADS33*, and *HsMADS145* was significantly higher compared to that of single flowers (Figure 8). The results indicated that the MADS-box genes were involved in flower organ development and could alter flower type by influencing the development of flower organs.

### 2.6. Analysis of MADS-Box Protein Interacting Network

We used TBtools to predict the potential functions of *MADS-box* genes in *H. syriacus*, and we constructed MADS protein interaction networks using Cytoscape 3.9.1 (accessed on 30 January 2023) (Figure 9). The results suggest that HsMADS8 may interact with SHORT VEGETATIVE PHASE (SVP), APETALA 2 (AP2), AGL24, and JOINTLESS (JOIN) proteins. The four proteins AGAMOUS (AG), SEP3, AGL11, and MADS4 may interact with each other. In addition, HsMADS8 and HsMADS10 proteins, which were homologous to APETALA 1 (AP1), had the highest correlation with other proteins.

## 3. Discussion

### 3.1. Genome-Wide Detection and Evolution of HsMADSs in H. syriacus

Using comparative genomic and phylogenetic analysis, 163 MADS-box genes in *H. syriacus* were identified and separated into five subfamilies. We found that the number of HsMADSs was smaller than that in *Gossypium hirsutum* (*n* = 207) [29], but larger than that in *Arabidopsis* (*n* = 102) [5], *G. max* (*n* = 106) [19], *G. raimondii* (*n* = 147) [29], *Malus domestica* (*n* = 146) [30], *Ananas comosus* (*n* = 44) [17], and *Citrullus lanatus* (*n* = 39) [31]. The large quantity differences between species may be caused by gene duplication events [32]. Furthermore, there were 134 type II genes in *H. syriacus*, far more than the 46 type II genes in *A. thaliana*. This indicated that the type II MADS-box genes of *H. syriacus* experienced a higher replication rate and/or a lower gene loss rate after replication, which was similar to the results of *G. max* [19]. We speculate that the presence of additional type II MADS-box genes in the *H. syriacus* genome indicates their high importance in the complex transcriptional regulation. In addition, there were more type II genes homologous to *Arabidopsis* than type I genes in *H. syriacus* (Figure 2b). Previous studies have indicated that the evolution of type II genes primarily resulted from genome-wide replication, while the evolution of type I MADS-box genes occurred through more recent and smaller-scale duplications [17]. It was relatively easier to identify orthologs of type II MADS-box genes from *Arabidopsis* compared to type I in *H. syriacus* and *G. raimondii*. This is due to localized duplications within each genus, which are more common for type I genes [33].

A total of 10 motifs were identified in *H. syriacus*, and the analysis of conservative motif distribution and the prediction of the proteins’ 3D structure showed that the HsMADS proteins were highly conserved in structure. The study of the intron distribution pattern of *H. syriacus* revealed that the type I (Mα, Mβ, and Mγ) MADS-box genes lacked introns or only had one intron, and had a very simple gene structure (Figure 3 and Figure 4). However, the gene structure of type II genes (MIKC and M*δ*) seemed more complex, and they included up to 11 exons and extra domains. Genes containing many introns are considered conservative [30], so type I genes may be less conservative than type II MADS-box genes. In addition, based on the phylogenetic tree, type II MADS-box genes of *H. syriacus* were more closely related to *A. thaliana* than type I. Meanwhile, the result of multiple sequence alignment showed that the Mα group was less conservative. This indicated that type I genes evolved faster than type II genes, which was consistent with the previous studies of *A. thaliana* and *Z. mays* [3,18].

We also identified 10 important *cis*-acting elements of the MADS-box family, most of which were related to plant hormones. Some promoter elements (ABRE, GARE-motif, p-box, TGAGG-motif, and CGTGA-motif) were enriched multiple times in the promoter region of the HsMADS genes, illustrating that the HsMADS genes may be involved in regulating multiple hormone responses. Meristem response elements were mainly identified in type II genes, indicating the role of the type II MADS-box genes in meristem identity and floral organ identity, which was in accordance with *Callicarpa americana* [34].

### 3.2. Functional Prediction of HsMADSs in H. syriacus

Understanding the gene expression is critical for comprehending the molecular mechanism of biological development [35]. MADS-box genes are considered to be connected with floral organ identity. Previous studies have shown that members of the *AOAMOUS* (*AG*) and *APETALA1* (*AP1*) genes, as observed in *Arabidopsis* [5], *Solanum lycopersicum* [36], *G. raimondii* [29], *Citrullus lanatus* [31], and *G. max* [19], are primarily expressed in flowers, fruits, and floral buds, respectively. Meanwhile, *AG* inhibits the activity of class A genes when regulating the identity of stamens and carpels [37]. In this study, the expression of *HsMADS48*, *HsMADS143*, and *HsMADS161* which were homologous to *Arabidopsis AG* in double flowers, was significantly lower than that in single flowers, while the expression of *HsMADS8*, which was homologous to *Arabidopsis AP1*, showed the opposite pattern. We speculate that the formation of double flowers likely requires the regulation of *AG* and *AP1*, as observed in previous studies on *Tricyrtis macranthopsis* [37] and *Lagerstroemia speciosa* [20,38]. In rounds 3 and 4, the *AP1* that controls petal formation was overexpressed, and the *AG* that controls the development of stamens and pistils was downregulated, causing the petalization of stamens [37,39]. In the present study, *HsMADS69*, which was homologous to *SOC1*, and *HsMADS33*, which was homologous to *MADS3*, were expressed at higher levels in petaloid stamens compared to normal stamens, which is in alignment with previous research conducted on *Lilium* [40]. *MADS3* might redundantly regulate the identity of floral organs with *MADS6* genes [41]. Therefore, we speculate that *HsMADS69* and *HsMADS33* also regulate the formation of the double flower by affecting the development of floral organs.

Protein–protein interactions are the key for MADS-box protein function [42]. In *Arabidopsis*, the interactions between the AG protein with three SEP proteins (SEP1, SEP2, and SEP3) and AGL6 are related to the termination of floral organ identity [43]. In this study, HsMADS143, HsMADS4 (SEP3), HsMADS10 (AGL11), and HsMADS145 (JOIN) interacted with each other, indicating that their interaction would affect the determination of floral organs. SVP, AGL24, and JOINTLESS belong to the SVP subfamily [44]. A previous study of tomato indicated that the flowering time and inflorescence determinacy were affected by JOINTLESS [45]. AP1 plays a role in promoting the identity of meristem, which is essential for the normal development of sepals and petals at the late stage of floral development [46]. In *H. syriacus*, HsMADS8 interacted with four proteins (AP2, AGL24, SVP, and JOINTLESS). Previous studies showed that all AP1 homologs in tomatoes failed to interact with SEP homologs, while *Arabidopsis* AP1 interacted with corresponding SEP homologs [42]. In this study, the HsMADS8 protein of *H. syriacus* also did not interact with SEP, which was similar to tomato. It has been proven that HsMADS8 protein is involved in fruit and ovule development [47]. Therefore, the differences in protein interaction patterns may lead to differences in fruit development among species.

## 4. Materials and Methods

### 4.1. Identification of MADS-Box Genes in H. syriacus

Arabidopsis MADS-box protein sequences were used as baits to search for the target sequence of *H. syriacus*. The sequences of MADS-box genes were gained from genome databases of *Arabidopsis* (TAIR, http://www.arabidopsis.org/, accessed on 11 November 2022) and *H. syriacus* (https://www.ncbi.nlm.nih.gov/, accessed on 11 November 2022). Then the candidate genes were screened again in NCBI by BLASTP searching (https://blast.ncbi.nlm.nih.gov/Blast.cgi, accessed on 14 November 2022). In the end, the MADS-box gene of *H. syriacus* were further verified based on the conservative domain; then the accuracy of candidate gene domains was checked manually using NCBI Batch CD-search (https://www.ncbi.nlm.nih.gov/Structure/bwrpsb/bwrpsb.cgi, accessed on 17 November 2022). The Expasy website (https://web.expasy.org/compute_pi/, accessed on 20 November 2022) was used to analyze the physicochemical properties of candidate genes, including isoelectric point (PI) and molecular weight (MW).

### 4.2. Phylogenetic Analysis and Multiple Sequence Alignment of HsMADS-Box Genes

To analyze the evolutionary relationships among HsMADS-box proteins, the multiple protein sequences of all 163 HsMADS-box proteins were aligned using MEGA11.0 (accessed on 25 November 2022) and Jalview software (accessed on 29 November 2022) [48]. Finally, the sequence of approximately 60 amino acid DNA-binding domains within these proteins was used for analysis.

A phylogenetic tree was constructed using *H. syriacus*, *A. thaliana*, and *G. raimondii* to obtain a clearer knowledge of the evolutionary relationship of the MADS-box gene family in *H. syriacus*. *G. raimondii* was selected because it has high-quality reference genome and a close relationship with *H. syriacus*. The phylogenetic tree was constructed by using the MEGA 11.0 program (accessed on 25 November 2022) with the maximum likelihood (ML) method and the relevant parameters (JTT + G and 1000 bootstrap replicates) [49]. Finally, the results of evolutionary relationship analysis were visualized by the iTOL website (http://itol.embl.de/, accessed on 9 December 2022).

### 4.3. Gene Structure and Conserved Motif Analysis of HsMADS-Box Genes

The conservative sequences were analyzed and visualized by the MEME tool (https://meme-suite.org/meme/tools/meme, accessed on 15 December 2022) and the TBtools v2.031 program (accessed on 27 November 2022) [50]. Exon–intron structures of the *HsMADS-box* genes were analyzed using TBtools with default parameters.

The Swiss-Model (https://swissmodel.expasy.org/, accessed on 25 December 2022) was used to forecast and construct the 3D structure of MADS-box proteins, and then SAVES (https://saves.mbi.ucla.edu/, accessed on 30 December 2022) was used to detect the feasibility of the 3D protein structure [51]. Finally, the Pymol software (accessed on 29 November 2022) was used to predict the RMSD value of the homologous model [52].

### 4.4. Analysis of cis-Acting Elements of HsMADS-Box Genes

TBtools was used to extract the 2000 bp upstream of the *HsMADS-box* genes transcription start site, and the types of *cis*-acting elements were analyzed on the PlantCARE website (http://bioinformatics.psb.ugent.be/webtools/plantcare/html/, accessed on 10 January 2023) and finally visualized using TBtools (accessed on 30 November 2022).

### 4.5. Expression Analysis of HsMADS-Box Gene

A total of 1 μg RNA was extracted from the flower bud samples of single flower *H. syriacus* “Albus single” (S), and double flower *H. syriacus* “Yupanlongzhu” (D) at two different developmental stages (0.3 cm and 0.7 cm), and named S-0.3, S-0.7, D-0.3, and D-0.7, respectively. The Nanodrop method was used to detect purity, concentration, and integrity of RNA [53]. Subsequently, double-stranded cDNA synthesis was performed, followed by purification of the cDNA fragment before end repair. Finally, the poly(A) tail was added to facilitate connection between the double-stranded cDNA and the Illumina end sequencing adapter (Wuhan, China). The AMPure XP beads were used to screen cDNA of approximately 370–420 bp, perform PCR amplification, and purify the product again to establish a cDNA sequencing library. Finally, the cDNA library was entrusted to Wuhan Metware Company for sequencing using Illumina HiSeqTM 4000 (Wuhan, China). The Fragments Per Kilobase of exon model per Million mapped fragments (FPKM) were used to evaluate the transcript abundance (Appendix A). The heatmap tool in TBtools was used to visualize the expression of nine HsMADSs in two cultivars.

Quantitative real time polymerase chain reaction (qRT-PCR) was accomplished using Thermal Cycler Dice Real Time System II (Takara, Osaka, Japan). The PrimerQuest website (https://sg.idtdna.com/pages/tools/primerquest, accessed on 20 April 2023) was used to design fluorescent primers (Appendix A). The RH8 (LOC120135994) gene was amplified as the internal control to normalize the amount of the template [54]. The normalized (2^−∆∆Ct^) value was used to calculate the relative expression [55].

### 4.6. Functional Protein Association Networks of MADS-Box

The complete set of representative transcriptional protein sequences of *H. syriacus* was obtained from NCBI. The set of representative protein sequences and protein interaction network relationships of *Arabidopsis* were obtained from String-db (https://cn.string-db.org/, accessed on 30 January 2023), and the prediction of the protein interaction network of *H. syriacus* was completed by TBtools. Finally, Cytoscape and Betweenness Centrality (BC) were used for visualization and network construction [56].

## 5. Conclusions

In summary, we identified 163 MADS-box genes in *H. syriacus* and divided them into type I and type II subfamilies. The prediction of the proteins’ 3D structure showed that the HsMADS proteins were highly conserved in structure. Type I MADS-box proteins were less conservative and further linked to *A. thaliana* than type II MADS-box proteins. The type II MADS-box genes had far more introns compared to type I, indicating that type I MADS-box genes evolved faster than type II. The *cis*-acting elements of the *HsMADS-box* gene promoter mainly responded to plant hormones, including AUX, MeJA, GA, and ABA. The distribution of meristem response elements highlighted the function of the type II MADS-box genes in meristem identity and floral organ identity. In addition, the expression of *HsMADS8*, *HsMADS33*, *HsMADS48*, *HsMADS69*, *HsMADS111*, *HsMADS138*, *HsMADS143*, *HsMADS145*, and *HsMADS161* was significantly different between single and double flowers, indicating that they played a critical role in regulating the development of flower organs. The analysis of protein interaction networks also indicated that HsMADS8 might play a core role in inflorescence determinacy and flowering time. These findings provide insights into the formation of double petal flowers in *H. syriacus* and potential candidate genes for its breeding.

## Figures and Tables

**Figure 1 ijms-25-00406-f001:**
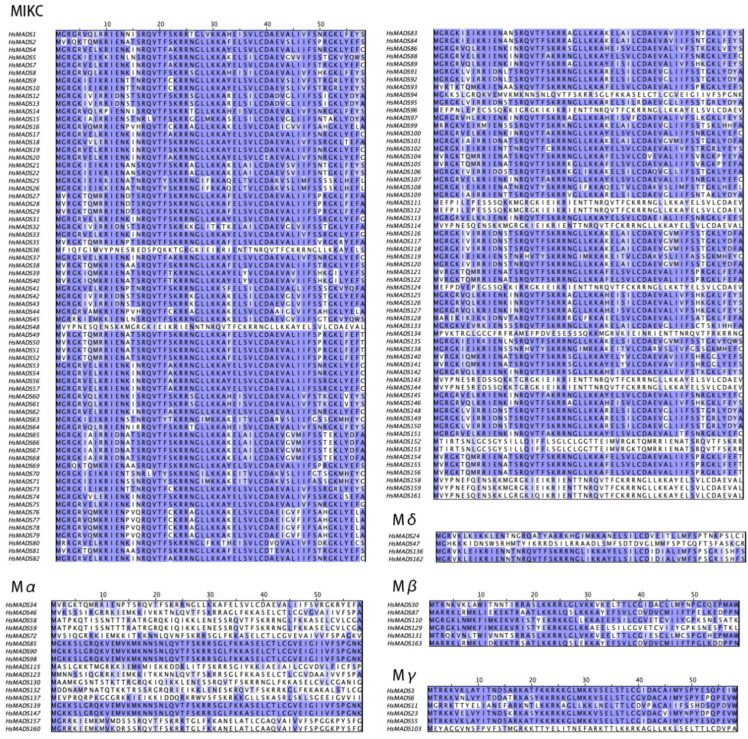
Multiple sequence alignment of the MADS domain from HsMADSs. Purple indicates a highly conserved region.

**Figure 2 ijms-25-00406-f002:**
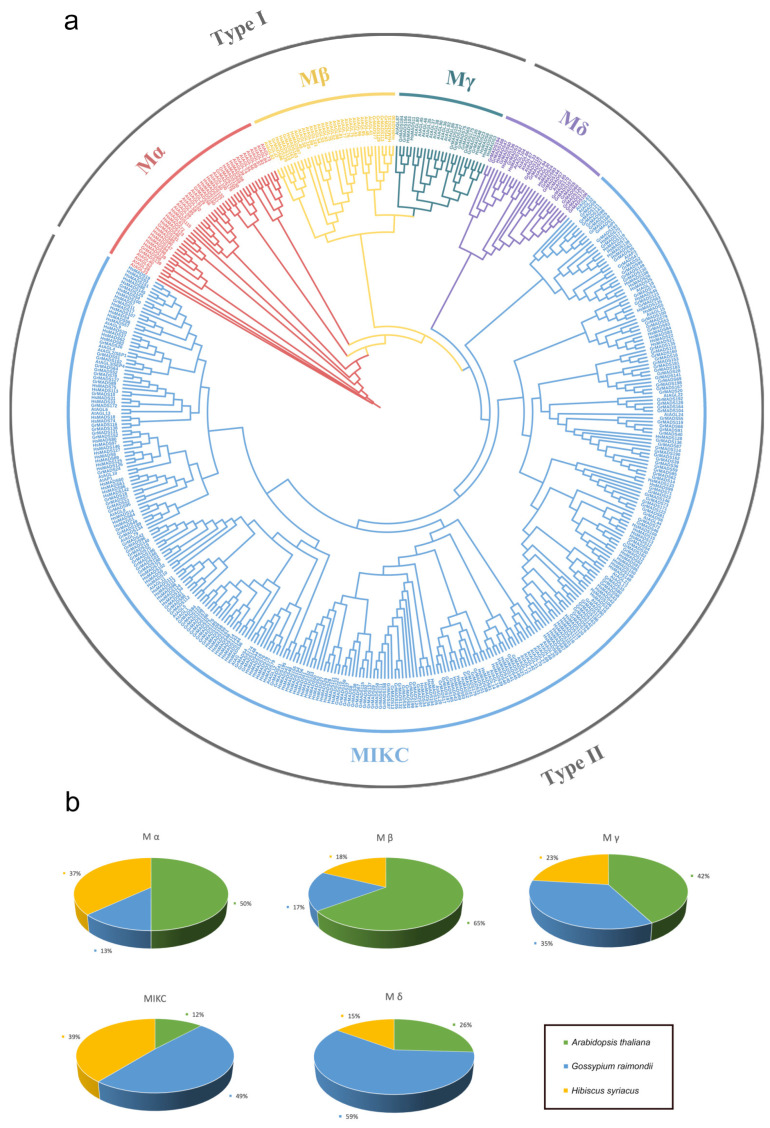
Phylogenetic tree of MADS-box protein sequences in *Arabidopsis thaliana, Gossypium raimondii*, and *Hibiscus syriacus*. (**a**) Phylogenetic analysis of 464 MADS-box protein sequences was divided into five subfamilies: MIKC, M*δ*, Mα, Mβ, and Mγ. (**b**) The percentage of MADS-box genes of three species in each subfamily. The yellow part indicates the proportion of MADS-box genes in *Hibiscus syriacus.* The green part indicates the proportion of MADS-box genes in *A. thaliana.* The blue part indicates the proportion of MADS-box genes in *G. raimondii*.

**Figure 3 ijms-25-00406-f003:**
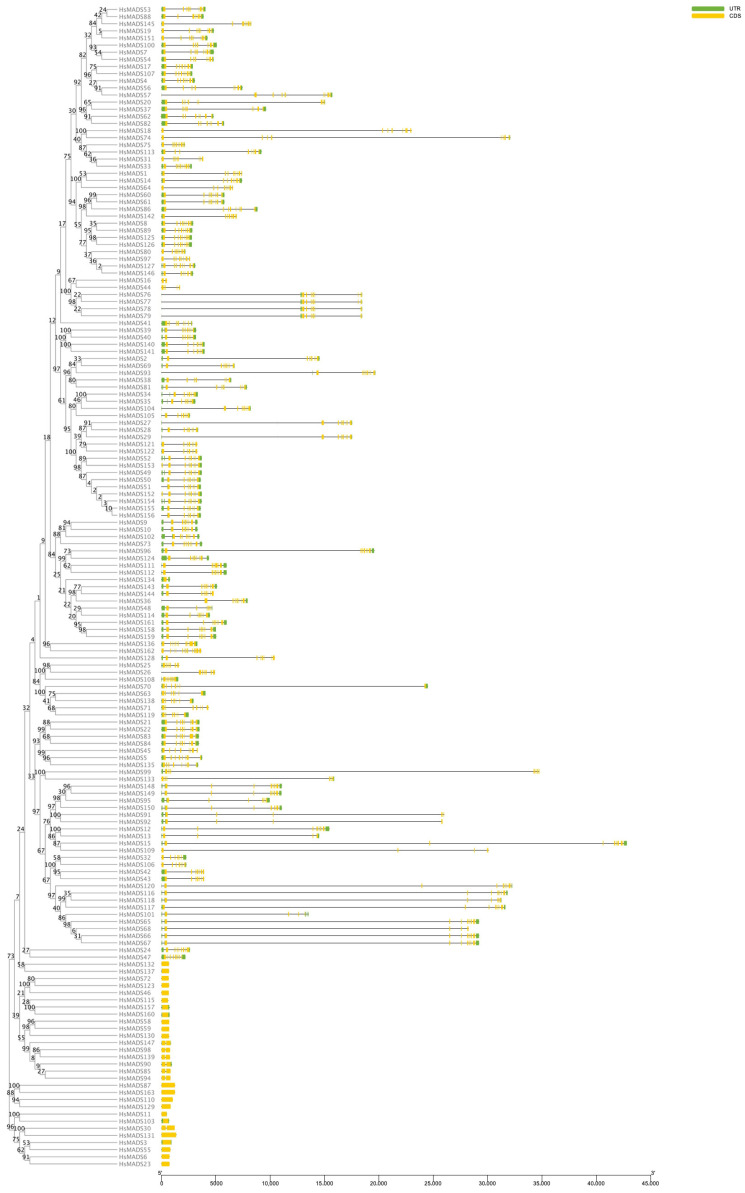
Gene structure of MADS-box genes in *Hibiscus syriacus*. Black lines indicate introns, green boxes indicate UTR, and yellow boxes indicate CDS.

**Figure 4 ijms-25-00406-f004:**
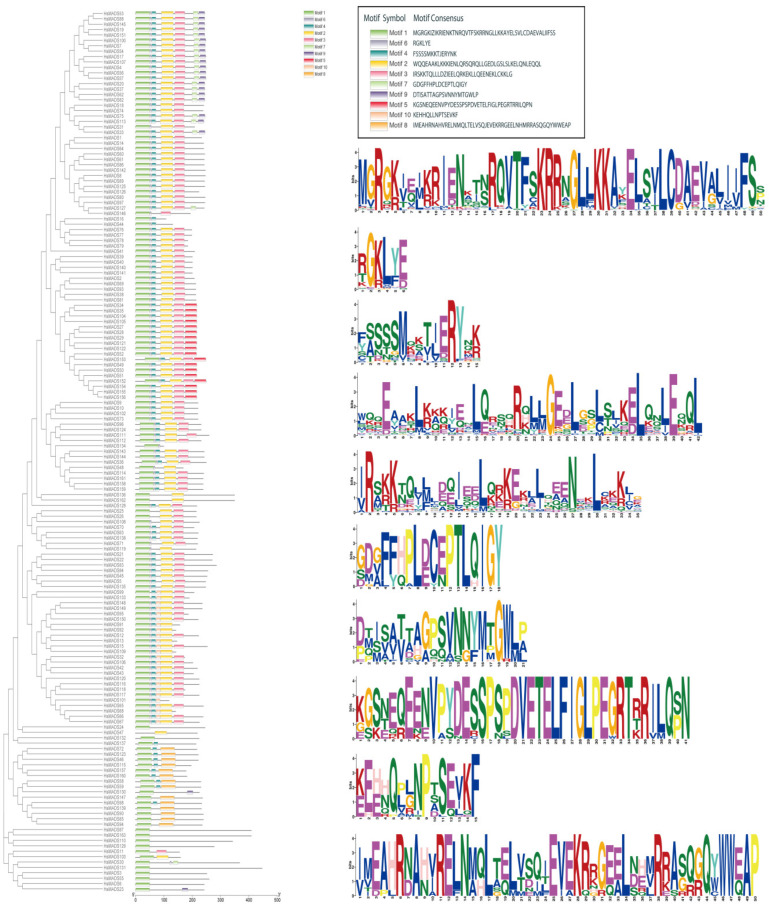
Motif composition of MADS-box proteins in *Hibiscus syriacus*. Details of ten motifs are shown.

**Figure 5 ijms-25-00406-f005:**
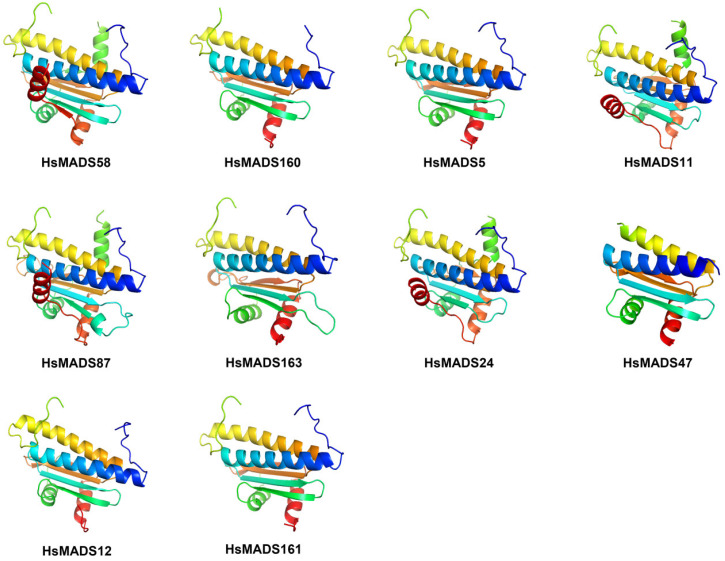
The three-dimensional structural mode of the HsMADS protein. HsMADS58 and HsMADS160 belong to the Mα subfamily, HsMADS87 and HsMADS11 belong to the Mβ subfamily, HsMADS5 and HsMADS163 belong to the Mγ subfamily, HsMADS24 and HsMADS47 belong to the Mδ subfamily, and HsMADS12 and HsMADS161 belong to the MIKC subfamily.

**Figure 6 ijms-25-00406-f006:**
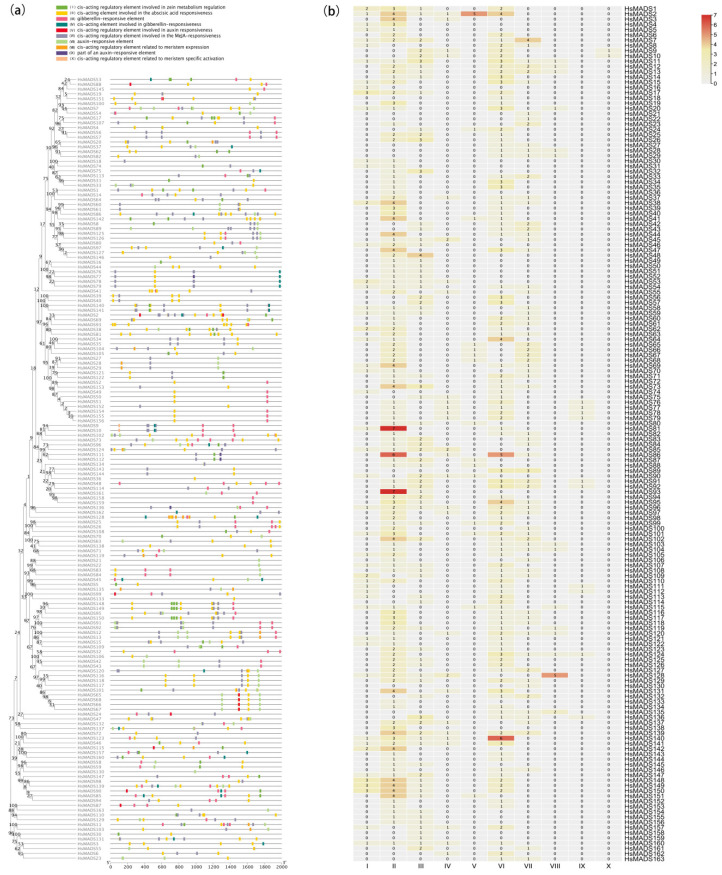
The *cis*-acting elements on putative promoters of HsMADS genes. (**a**) The distribution of ten *cis*-acting elements of HsMADS genes. A phylogenetic tree of HsMADS genes was constructed via MEGA11.0 using the Neighbor–Joining (NJ) method and 1000 bootstraps. (**b**) Number of *cis*-acting elements of *HsMADS-box* genes. Arabic numerals represent the number of different types of elements in *HsMADS* genes. A total of ten *cis*-acting elements were investigated, including: (I) *cis*-acting regulatory element involved in zein metabolism regulation, (II) *cis*-acting element involved in the abscisic acid responsiveness, (III) gibberellin-responsive element, (IV) *cis*-acting element involved in gibberellin-responsiveness, (V) *cis*-acting regulatory element involved in auxin responsiveness, (VI) *cis*-acting regulatory element involved in the MeJA-responsiveness, (VII) auxin-responsive element, (VIII) *cis*-acting regulatory element related to meristem expression, (IX) part of an auxin-responsive element, (X) *cis*-acting regulatory element related to meristem specific activation.

**Figure 7 ijms-25-00406-f007:**
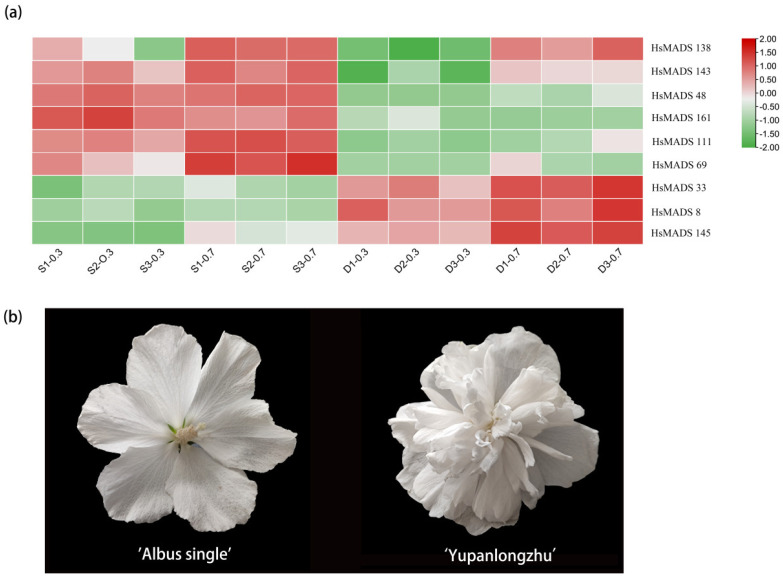
Expression of MADS-box genes in *Hibiscus syriacus*. (**a**) Heatmap of bud expression data in *H. syriacus* “Albus single” (S), and *H. syriacus* “Yupanlongzhu” (D) at two different stages, and named S-0.3, S-0.7, D-0.3, and D-0.7, respectively. The red color represents higher expression levels and green represents lower expression levels. (**b**) Phenotypes of two *Hibiscus syriacus* species. “Yupanlongzhu” is a double flower, “Albus single” is a single flower.

**Figure 8 ijms-25-00406-f008:**
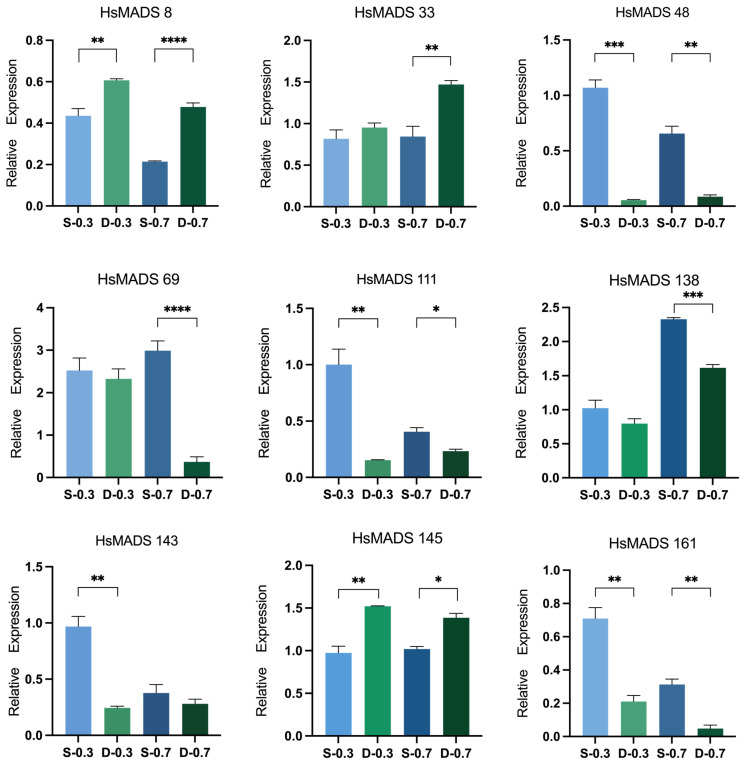
Relative expression of *HsMADS8*, *HsMADS33*, *HsMADS48*, *HsMADS69*, *HsMADS111*, *HsMADS138*, *HsMADS143*, *HsMADS145,* and *HsMADS161* in response to floral organ development according to RT-qPCR analysis. Error bars indicate the standard deviation of biological replicates. * *p* < 0.05, ** *p* < 0.01, *** *p* < 0.001, and **** *p* < 0.0001.

**Figure 9 ijms-25-00406-f009:**
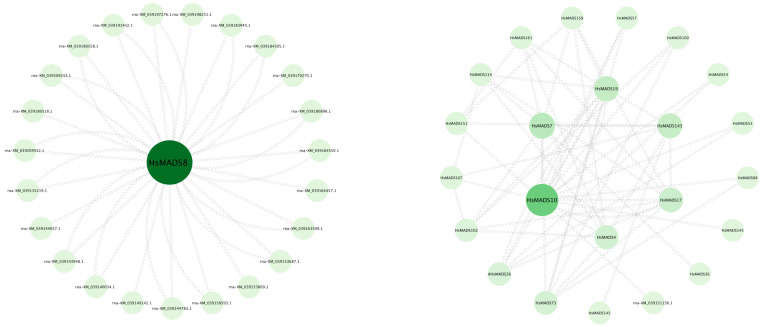
Protein interaction network of HsMADSs proteins in *Hibiscus syriacus* constructed by the orthologues in Arabidopsis.

## Data Availability

The datasets presented in this study can be found in online repositories. The names of the repository/repositories and accession number (s) can be found below: https://www.ncbi.nlm.nih.gov/, PRJNA1047745, accessed on 7 December 2023.

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
