# Peer review of "Genome-Wide Analysis of the MADS-Box Gene Family in Hibiscus syriacus and Their Role in Floral Organ Development"

_ijms, 2023, doi:10.3390/ijms25010406_

Round 1

Reviewer 1 Report

Comments and Suggestions for Authors

The authors perform a genome-wide characterization of MADS-box proteins in Hibiscus syriacus, and then investigate gene structure, phylogenetics, cis-acting elements, three dimensional structure, gene expression, and protein interaction to identify candidate MADS-box genes in that species.

Up to now, in hibiscus spp there are no specific information available about the MADS-box gene family, a family of transcription factors that play a crucial role in the regulation of various developmental processes in plants, including flower development. The work is therefore interesting because this research added important information about MADS-box genes in that ornamental species and I think the manuscript worth to be published, however, some aspects of the manuscript need to be clarified and better organized, thus I do not feel that in the present form the manuscript is ready for publication in IJMS. In particular the description of methods should be more accurate, with some more details on the method to identify the MADS-box gene family in the H.syriacus genome.

My concerns about the manuscripts are listed below:

Introduction

Line 54 Add the references of a recent review on MADS box genes family  in plants Castelán-Muñoz, Natalia, et al. "MADS-box genes are key components of genetic regulatory networks involved in abiotic stress and plastic developmental responses in plants." Frontiers in Plant Science 10 (2019): 853.

Line 65 Although it would be necessary, it lacks the description of the main botanical characteristics of the species under consideration: eg mating system, ploidy, chromosome number, genome size, which are useful for the reader who is unfamiliar with the species object of the research.

Line 67. Several studies of the molecular mechanism underlying double flowers exist, you should cite some of them

Results

Lines 79-89 The authors identify 277 candidate genes: did they use a method to reduce redundancy? For instance were redundant sequences with the same chromosome locus remove to ensure that the candidate genes were mapped to unique loci in their respective genomes?

Table S1 It is protein sequence length, isn’t it? So please write “size (aa)”

Line 101 Specify, at least in the materials and methods section why you choose G. raimondii to investigate the evolutionary relationships of MADS box family members in H. syriacus

Lines 99 - 115 and Figure 2. In my opinion, It would be interesting to separate the two types of MADS box genes (type  and II) and obtain phylogenetic trees of type I and type II MADS-box genes.

Lines 141-142 On what basis you selected two MADS-box genes from each subfamily to predict the 3-D structure of….please explain.

Lines 193-194 six MADS box genes were randomly selected to conduct qRT-PCR analyses, but maybe “randomly” is not the right word as they are all the differently expressed genes (lines 181-186) except one (HsMADS 138).

Materials and methods

Lines 295 Move Gossypium raimondii in the 4.2 section as here you use the A. thaliana to search for MADS box genes in H. syriacus and explain here in section 4.2 why you use Gossypium raimondii as model species.

Lines 297 please rephrase specifing that you use Arabidopsis MADS-box protein sequences as queries to perform a BLASTP search against the H. syriacus genome (or proteome). Also add the date of database access and data acquisition. Please do the same in lines 303 and 319/320

Are H. syriacus genome annotation data available? If yes, you should add chromosome localization of the MADS-Box gene family

Line 385. Data availability: Has the raw transcriptome dataset been submitted to the GenBank, so that data are available to the public?

Reviewer 2 Report

Comments and Suggestions for Authors

In this manuscript, the authors succinctly conveyed that MADS-box genes serve as crucial regulators influencing nearly every facet of plant reproductive development. Their significant contributions are particularly notable in the control of flowering time, shaping inflorescence architecture, determining floral organ identity, and orchestrating seed development processes. Excellent work, indeed.

However, I have a few observations to improve this manuscript, as follows:

a. In Figure 2, the scientific names are not italic. Please correct those.

b. In the Figure 5 legend, the figure is not self-explanatory. Please explain the figure in brief.

c. I haven't seen any explanation in fig. 7 and discussion parts on how reproductive organs are modifying into petal in Hibiscus. Could you add some of your mechanism thoughts related to MADS-box? That will make sense.

Other portions are fine to me. Looking forward to seeing your revised manuscript.

Round 2

Reviewer 2 Report

Comments and Suggestions for Authors

Well revised. Thank you.
